# CFD Analysis of Sine Baffles on Flow Mixing and Power Consumption in Stirred Tank

**Shuiqing Zhou** [1,2,*], **Qizhi Yang** [1,2], **Laifa Lu** [1,2], **Ding Xia** [1,2], **Weitao Zhang** [1,2] **and Hao Yan** [3]

1. College of Mechanical Engineering, Zhejiang University of Technology, Hangzhou 310023, China; yqz2539552237@163.com (Q.Y.); llf1468784036@163.com (L.L.); 2112102509@zjut.edu.cn (D.X.); tao1039550784@163.com (W.Z.)
2. Innovation Research Institute of Shengzhou, Zhejiang University of Technology, Shengzhou 312400, China
3. College of Mechanical Engineering, Hefei University of Technology, Hefei 230009, China; yanying0708@126.com
* Correspondence: zsqwh986@zjut.edu.cn

**Abstract:** In order to enhance the fluid mixing in the stirred tank and reduce the power consumption under the condition of full baffle, a sinusoidal sawtooth baffle was established in the present study. Based on the Eulerian–Eulerian method, a numerical simulation of the mixed flow in the stirred tank was conducted, and the reliability of the simulation method was verified by means of PIV experiments. The different structural characteristics of a standard baffle and the sine baffle were compared, to explore the effect of the modified baffle on flow mixing and power consumption in the tank. The outcomes indicate that the sinusoidal sawtooth structure had the effect of reducing drag and shunting, which could lessen the backflow on the backside of the baffle, strengthen the intensity of fluid turbulence and strain rate, improve the uniformity of particle distribution, and significantly lower the power consumption. When the relative tooth height was 0.333 and the relative tooth width was 0.028, the power consumption was reduced by 11.7%.

**Keywords:** sine baffle; stirred tank; CFD; power consumption

## 1. Introduction

As significant process equipment for ensuring production, stirred tanks are mainly used for operation units such as mixing, dissolution, crystallization, extraction, mass transfer, and heat-transfer in various industries, including medicine, dye, food, metallurgy, wastewater treatment, and synthetic materials [1–3]. The flow information of the fluid is the basis for investigating the energy, quality, and reaction process in the vessel, and it determines the output and energy consumption of products in industrial production. The stirring effect depends on the type of impeller, the physical properties of the medium, and the baffle structure.

In general, when dealing with low viscosity materials, the rotation speed increases and the flow is in a turbulent state. The centrifugal force and the inner wall of the stirred tank work in conjunction to enhance the circular motion of the medium and form a vortex on the free liquid surface, thereby weakening the mixing effect (Figure 1). Through the baffle settings, most of the tangential flow caused by the rotation of the impeller is converted into more efficient axial and radial flow, which can minimize the formation of the central vortex and effectively disrupt the flow of circulating fluid [4]. Researchers in the relevant field have conducted a number of experiments and numerical simulations [5]. Ying F et al. [6] used PIV experiments to compare and analyze the baffled and non-baffled stirred tank. The addition of baffles increased the convective circulation of the main body and created favorable conditions for the diffusion of a small-scale structure under the stirred tank. Atibeni R et al. [7] compared the effects of different width baffles and up and down triangular baffles on the suspended particle stirred tank. Kamla Y et al. [8] evaluated the

power consumption in the Rushton turbine agitator at different baffle inclinations. After conducting a dynamic analysis of turbulent flow in a stirred tank, Ammar M et al. [9] found that the power number strongly depended on the baffle length. Lin et al. [1] investigated the effect of the baffle structure and groove size on dimethyl fumarate spherical agglomerates and found that with the increase of baffle width, the strain rate and turbulent dissipation rate increased, rendering an increase in the collision rate of aggregates. Vitor et al. [10] compared vertical tube baffles with traditional jacket and helical coil heat-transfer methods; the former eliminated eddy currents and had higher heat exchange efficiency. Several scholars have described the effect of the special-shaped baffle design. Soliman et al. [11] analyzed the heat and mass transfer behavior of a novel heterogeneous stirred tank reactor with serpentine baffles, which could effectively improve the selectivity and yield of the reactor during the exothermic liquid–solid diffusion-controlled catalytic reaction process. Shen et al. [12] installed a "V"-shaped horizontal baffle at the height of the impeller, and such a structure could reduce energy consumption and improve the mixing effect during the mixing process of liquid–liquid two-phase flow. Bukhari et al. [13] evaluated the coefficient of variation (COV) of a fractal baffled stirred tank, which improved the mixing efficiency compared with ordinary baffles. In a six-blade Rushton turbine stirred tank, Foukrach et al. [14] quantified and characterized the effect of baffle shape on the flow field and power consumption and found that increasing the baffle curvature could reduce power consumption.

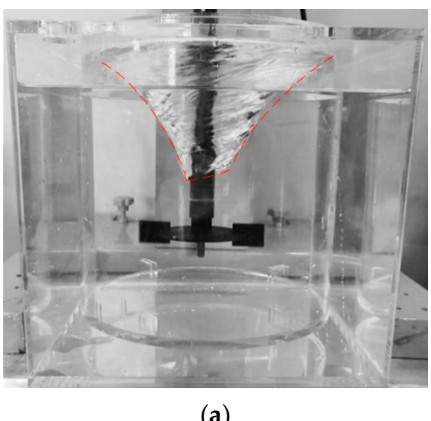 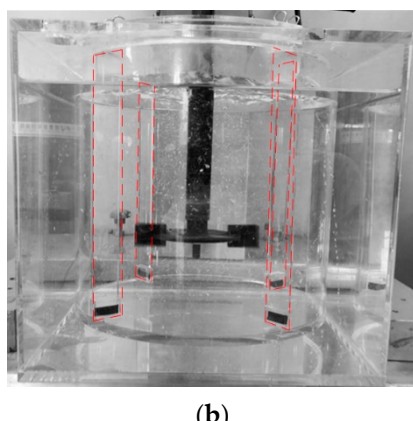

(**a**)            (**b**)

**Figure 1.** The comparison of the free liquid surface with and without baffles at 450 rpm. (**a**) Without baffles; (**b**) standard baffles.

Scholars have conducted experimental and model studies on the hydrodynamic characteristics of stirred tanks, such as stirring speed and turbulence intensity. On the premise of not changing the shape and working conditions of the impeller, the baffle plate provides an effective method for strengthening the stirring. However, the standard baffle structure affects the stirring speed and mixing uniformity while also increasing the energy consumption. Despite the fact that several scholars having explored the sawtooth, there is a scarcity of reports on the influence mechanism of the sawtooth shape, the relative tooth height, and the relative tooth width on the stirring flow field. In the present study, numerical simulations and PIV experiments were performed to investigate the effects of sinusoidal sawtooth baffles on the fluid velocity field distribution, particle concentration distribution, and power consumption in a stirred tank equipped with a six-blade turboprop (PBTD).

## 2. Research Objectives

### 2.1. Stirring Device

The stirring device was a flat-bottomed cylindrical vessel, and the impeller was a 45° six-blade turboprop (PBTD). The standard baffle agitator is shown in Figure 2. In the

present study, the flow state in the stirred vessel was defined by the Reynolds number (*Re*), *Re* = 127,180.8.

$$Re = \frac{\rho_l N D^2}{\mu} \tag{1}$$

The parameters are shown in Table 1.

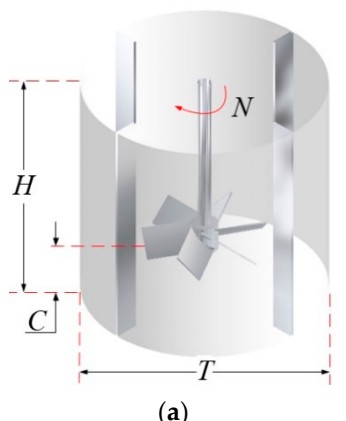
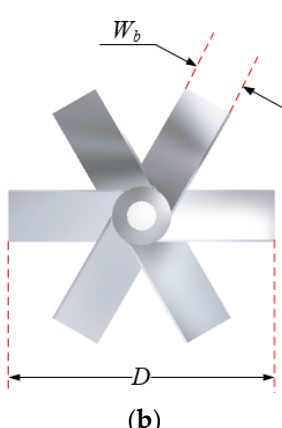

(**a**)    (**b**)

**Figure 2.** A schematic diagram of the stirring device. (**a**) The stirred tank; (**b**) PBTD.

**Table 1.** The stirring component dimensions.

| Element | Symbol | Value |
|---|---|---|
| Tank diameter | $T$ | 288 mm |
| Impeller diameter | $D$ | $T/2$ |
| Off-bottom clearance | $C$ | $T/4$ |
| Blade width | $W_b$ | 38 mm |

### 2.2. Sine Baffle Structure

The purpose of adding baffles is to change the structure of the flow field, enhance the strength of the fluid convection circulation, provide a stable free liquid surface, and then promote mixing. However, the baffle settings also greatly improve the stirring power consumption, and a circulation dead zone can easily form on the back side of the baffle, which reduces the uniformity of fluid mixing. Karcz et al. [15] explored the influence of the number, length, and installation method of the baffles on particle suspension characteristics and affirmed the superiority of non-standard baffles. As such, an ideal baffle structure can achieve a mixed effect of high quality and low power consumption. From the perspective of energy loss, Zhang et al. [16] analyzed the drag reduction mechanism of the sawtooth guide vanes, which promoted the fluid mixing in the pipe elbow and had a positive effect on reducing the energy dissipation at the leading edge of the guide vane. Sergio A. et al. [17] introduced U-shaped and V-shaped cut impellers, which improved the turbulent kinetic energy level and energy dispersion in the outflow area of the impeller and reduced the power consumption. Several studies [18,19] have shown that structural cutting was beneficial for breaking the mixing isolation zone, had a key impact on wake vortex and energy dissipation, was conducive to the effective mixing of fluids and medium diffusion, and had a positive effect on reducing power consumption. In the present study, based on the above-mentioned beneficial effects, the standard baffle in the stirred tank was edge-cut, and the profile curve of the cut was a sine function, with the shape thereof being determined by the relative tooth height ($A/W$) and relative tooth width ($\lambda/h$). In order to facilitate the calculation of the mathematical model and the demonstration of results, several structures similar to Figure 3 were taken as examples in this paper. Table 2 shows the characterization parameters of the sine baffle.

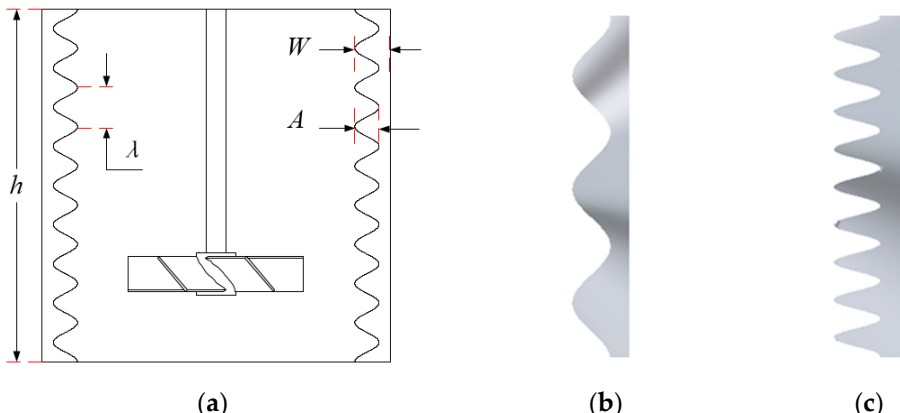

**Figure 3.** A schematic diagram of the stirring device with sine baffles. (**a**) Agigator, (**b**) $A/W = 0.333$, $\lambda/h = 0.104$, and (**c**) $A/W = 0.333$, $\lambda/h = 0.028$.

**Table 2.** The structural dimensions of bionic sine baffle.

| Element | Symbol | Value |
|---|---|---|
| Baffle width | $W$ | 30 mm |
| Baffle height | $h$ | 288 mm |
| Tooth height | $A$ | 10~20 mm |
| Tooth width | $\lambda$ | 8~40 mm |
| Baffle thickness | $\delta$ | 3 mm |

## 3. PIV Experimental Setup

In the present study, a 2D PIV system was used (Microvec Pte Ltd., Beijing, China), comprising a 532 nm Nd-Yag double-pulse laser (10 Hz, 500 mJ), a 6600 × 4400 pixel high-resolution CCD camera, a synchronizer, and commercial software Microvec-V3.6 composition. The experimental setup is shown in Figure 4a. An electromagnetic-phase locking device was used to ensure that the measurement phase was invariant, that is, a signal was triggered per shaft revolution and the blade angular position, image acquisition, and laser were synchronized. Due to the symmetry, only half of the flow field of the stirred tank was measured.

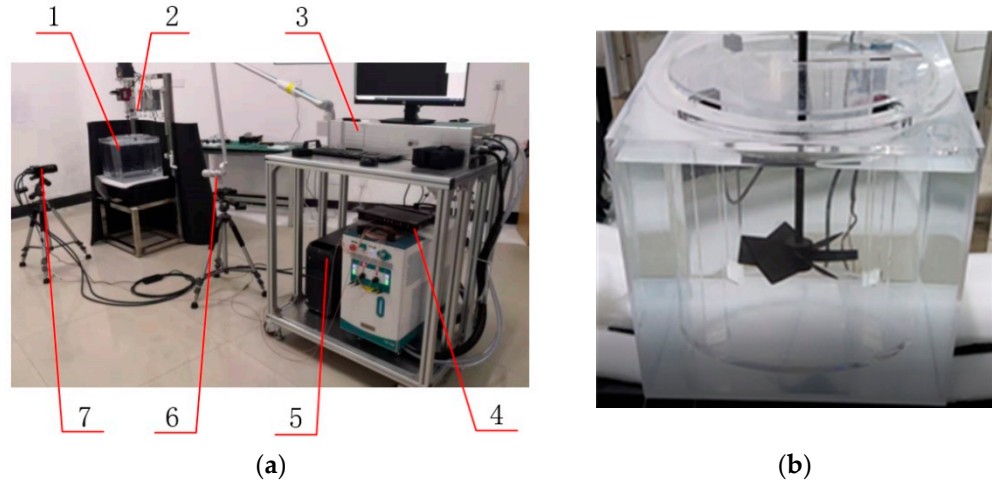

**Figure 4.** Experimental device. 1—stirring device, 2—electromagnetic phase locking device, 3—laser, 4—synchronizer, 5—computer, 6—cylindrical, and 7—CCD camera. (**a**) PIV device; (**b**) stirred tank.

The tracer particles used in the present experiment were hollow glass microbeads with a diameter of 10–20 μm and a density of 1030 kg/m³—close to the liquid density for

the better tracking of the liquid velocity. Additionally, the liquid–solid physical property parameters used in this experiment are shown in Table 3. In the correlation calculation process, the selected interpretation area was $64 \times 64$ pixels, and the overlapping area of every two adjacent interpretation areas was 50%. Taking into account the accuracy of the experimental data and the efficiency of image processing, the 500 pairs of images collected were processed with a time average to obtain the average flow field under each working condition.

**Table 3.** The physical parameters.

| Parameter | Symbol | Value |
|---|---|---|
| Liquid density | $\rho_L$ | 1150 kg/m$^3$ |
| Hydrodynamic viscosity | $\mu$ | 0.001 pa·s |
| Solid particle density | $\rho_S$ | 2485 kg/m$^3$ |
| Solid particle diameter | $d_S$ | 3 mm |
| Solid volume fraction | $\varphi_S$ | 5.2% |

The full baffle flat-bottomed cylindrical vessel was experimentally investigated, and the dimensions are shown in Table 1. A cylindrical stirred tank was placed in a square water tank to reduce the effect of light refraction at the cylindrical surface on the experiment. The impeller and shaft were black painted, to prevent the high-speed camera (CCD) from being damaged by strong laser reflection (Figure 4b).

## 4. Mathematical Model

Using computational fluid dynamics (CFD), researchers can obtain fundamental data on large numbers of samples in a simpler and more economical manner than with experiments. The current numerical simulation methods of multiphase flow include the Eulerian–Lagrangian method and the Eulerian–Eulerian method, the latter of which requires less calculation and is more suitable for the calculation of liquid–solid two-phase flow under high solid holdup.

### 4.1. Governing Equations

To simplify the calculations, the system was assumed to flow isothermally, the liquid was incompressible, and the particles were spherical with a single size. The model regarded the liquid–solid two phases as a continuous medium filled with flow field and mutual interaction, and the Reynolds-averaged mass and momentum conservation equations of each phase were solved. The governing equation [20] is:

Continuity Equation:

$$\frac{\partial}{\partial t}(\rho_i \varphi_i) + \nabla \cdot (\rho_i \varphi_i \vec{u}_i) = 0 \tag{2}$$

where i represents the liquid phase (*L*) or solid phase (*S*); $\rho$ is the density; $\varphi$ is the phase volume fraction; and $\vec{u}$ is the velocity vector.

The momentum conservation equation:

The momentum conservation equation for the liquid phase is:

$$\frac{\partial}{\partial t}(\varphi_L \rho_L \vec{u}_L) + \nabla \cdot (\varphi_L \rho_L \vec{u}_L \vec{u}_L) = -\varphi_L \nabla P + \varphi_L \nabla \cdot \bar{\bar{\tau}}_L + K(\vec{u}_S - \vec{u}_L) + \varphi_L \rho_L g \tag{3}$$

$$\bar{\bar{\tau}}_L = \varphi_L \mu_L (\nabla \vec{u}_L + \nabla \vec{u}_L{}^T) + \varphi_L (\lambda_L - \frac{2}{3}\mu_L) \nabla \cdot \vec{u}_L \tag{4}$$

The momentum conservation equation for the solid phase is:

$$\frac{\partial}{\partial t}(\varphi_S \rho_S \vec{u}_S) + \nabla \cdot (\varphi_S \rho_S \vec{u}_S \vec{u}_S) = -\varphi_S \nabla P - \nabla P_S + \nabla \cdot \bar{\bar{\tau}}_S + K(\vec{u}_L - \vec{u}_S) + \varphi_S \rho_S g \tag{5}$$

$$\bar{\bar{\tau}}_S = \varphi_S \mu_S (\nabla \vec{u}_S + \nabla \vec{u}_S{}^T) + \varphi_S (\lambda_S - \frac{2}{3}\mu_S) \nabla \cdot \vec{u}_S \tag{6}$$

$$p_S = \varphi_S \rho_S \theta_S + 2\rho_S(1 + e_{SS})\varphi_S{}^2 g_{0,SS}\theta_S \tag{7}$$

$$g_{0,SS} = \left[1 - \left(\frac{\varphi_S}{\varphi_{S,\max}}\right)^{1/3}\right]^{-1} \tag{8}$$

where $p$ is the thermodynamic pressure; $p_S$ is the solid phase pressure; $K$ is the drag coefficient between the solid and liquid phases, calculated from the resistance coefficient model; $\overline{\overline{\tau}}_L$ and $\overline{\overline{\tau}}_S$ are the liquid and solid phase shear stress tensor, respectively; $\theta s$ is the granular temperature for the solid phase; $\lambda_S$ and $\mu_S$ are the particle phase bulk viscosity and shear viscosity, respectively; $e_{SS}$ is the inelastic collision restitution coefficient of particles; $g_{0,SS}$ is the particle radial distribution function; and $\varphi_{S,\max}$ is the maximum particle packing.

Huilin et al. [21] further modified the continuous form of resistance in the Gidaspow model by using a smooth function, realized the continuous transition from low solid holdup to high solid holdup, and obtained Huilin–Gidaspow with a wider range of solid holdup in the model to calculate the momentum exchange coefficient:

$$K = (1 - \psi)K_E + \psi K_{WY} \tag{9}$$

where $\psi$ is a smooth function, and the expression is as follows:

$$\psi = \frac{arctan[150 \times 1.75(0.2 - \varphi_S)]}{\pi} + 0.5 \tag{10}$$

$$K_E = 150\frac{(1 - \varphi_L)^2\mu_l}{(\varphi_L d_S)^2} + 1.75\frac{\rho_l(1 - \varphi_L)\left|\vec{u}_L - \vec{u}_S\right|}{\varphi_L d_S} \quad \varphi_L \leq 0.8 \tag{11}$$

$$K_{WY} = \frac{3}{4}C_d\frac{\rho_L(1 - \varphi_L)\left|\vec{u}_L - \vec{u}_S\right|}{d_S}\varphi_L^{-2.65} \quad \varphi_L > 0.8 \tag{12}$$

where the drag coefficient $C_d$ is:

$$C_d = \frac{24}{Re_S}\left[1 + 0.15Re_S{}^{0.687}\right] \tag{13}$$

In the formula, $Re_s$ is the particle Reynolds number, which is calculated by the following formula:

$$Re_S = \frac{\rho_L d_S\left|\vec{u}_S - \vec{u}_L\right|}{\mu_L} \tag{14}$$

Montante [22] found that the mixture model is the most proper turbulence model. For this reason, both the turbulent kinetic energy and dissipation rate of the liquid phase were calculated using the mixture $k$-$\varepsilon$ turbulence model where the two phases are assumed to share the same $k$ and $\varepsilon$. The $k$ and $\varepsilon$ equations describing this model are as follows [23]:

$$\frac{\partial}{\partial t}(\rho_m k) + \nabla \cdot (\rho_m \vec{u}_m k) = \nabla \cdot \left(\mu_m + \frac{\mu_{t,m}}{\sigma_k}\nabla k\right) + G_{k,m} - \rho_m \varepsilon \tag{15}$$

$$\frac{\partial}{\partial t}(\rho_m \varepsilon) + \nabla \cdot (\rho_m \vec{u}_m \varepsilon) = \nabla \cdot \left(\frac{\mu_{t,m}}{\sigma_\varepsilon}\nabla \varepsilon\right) + \frac{\varepsilon}{k}(C_{1\varepsilon}G_{k,m} - C_{2\varepsilon}\rho_m \varepsilon) \tag{16}$$

where the mixture density $\rho_m$, the mixture viscosity $\mu_m$ and the mixture velocity $\vec{u}_m$ are calculated from:

$$\rho_m = \sum_{i=1}^{z}\varphi_i\rho_i \tag{17}$$

$$\mu_m = \sum_{i=1}^{z}\varphi_i\mu_i \tag{18}$$

$$\vec{u}_m = \frac{\sum\limits_{i=1}^{z} \varphi_i \rho_i \vec{u}_i}{\sum\limits_{i=1}^{z} \varphi_i \rho_i} \tag{19}$$

The turbulent viscosity of the mixture can be calculated from the above equations:

$$\mu_{t,m} = \rho_m C_\mu \frac{k^2}{\varepsilon} \tag{20}$$

and the turbulence generation $G_{k,m}$ is:

$$G_{k,m} = \mu_{t,m} (\nabla \vec{u}_m + \nabla \vec{u}_m^T) : \nabla \vec{u}_m \tag{21}$$

The coefficients of the turbulent model involved in the above equations are $C_{1\varepsilon} = 1.44$, $C_{2\varepsilon} = 1.92$, $C_\mu = 0.09$, $\sigma_k = 1.0$, and $\sigma_\varepsilon = 1.3$, respectively.

### 4.2. Numerical Simulation Details

In the present study, the commercial CFD double-precision solver FLUENT was used to solve the liquid–solid two-phase governing equation. The Eulerian model was chosen as the multiphase model of the mixed flow and the pressure-based transient implicit solution method. To realize the coupling of pressure and velocity, an independent SIMPLEC algorithm was adopted. The convection terms were discretized in the second-order upwind scheme, and all the diffusion terms were in the central difference format. Additionally, the temporal term was discretized by the second order implicit algorithm.

The sliding grid (SG) algorithm was applied to the simulation of the impeller rotation with the set time step and the time-step stop criterion of 0.005 s and $10^{-4}$, respectively, and the Courant number was restricted less than 1. As far as the SG simulations were concerned, 50 s was enough to satisfy the stable flow field conditions. To model the near wall regions, the standard wall function was used. The logarithmic law for the mean velocity is known to be valid for 30 < y+ < 300. The y+ values calculated in this study were within this range. All geometry walls were assumed to be the no slip boundary conditions, except that the top surface adopts free slip condition to simulate the free surface. Under the initial conditions, the solid particles were settled uniformly at the bottom of the tank, while the liquid was kept stationary in the remaining space of the tank. The mesh of the impeller and baffle was refined to capture better flow field details [14], and the independence of the mesh was verified. If the number of grid cells was too small, the precision of the numerical simulation results would be low, and there would be a large deviation from the experimental results. At the same time, if the number of grid cells was too large, not only would there be little improvement in the calculation accuracy, the calculation cost would also be increased. The specifications of the standard baffle agitated tank experimental device used in the present study were consistent with those in a previous study [18], and the grid independence verification results were referred to. When the number of grid cells was 800 k, the average velocity exhibited a similar trend as the experiment, but the agreement was lower. When the number of grid cells was between 1200 k and 1800 k, the calculation requirements and accuracy could be met (Figure 5). Since the research object of this paper was the baffle, it was necessary to refine the mesh at the baffle. The number of grid cells of the sinusoidal sawtooth baffle stirred tank was selected from 2200 k to 2800 k.

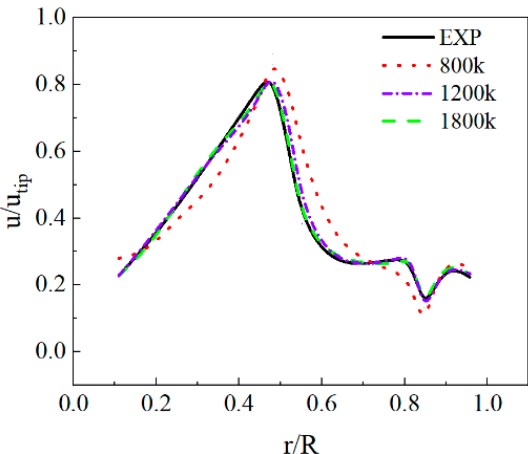

**Figure 5.** The average velocity of fluid with different mesh numbers compared with the experiment (y = 0.25 h).

*4.3. Numerical Model Verification*

The reliability of the numerical model was verified by means of PIV experiments. Due to the high particle volume fraction and impeller speed, the application of optical measurement technology would be affected if the number of particles was too large. Therefore, the experimental and simulated velocity vectors were compared at 240 rpm (*Re* = 95,385.6) and solids volume fraction of 2%. Figure 6 shows that the liquid phase mean velocity fields of the simulations and PIV experiments were similar. From r/R = 0.3 to r/R = 0.4, the main circulation vortex was formed near the blade tip. At the same time, the fluid flowed out obliquely downward from the tip at a maximum speed of $u_{tip}$ = 1.8 m/s. After contacting the bottom of the kettle, 80% of the fluid upwards participated in the main circulation of the entire stirred tank, and the rest formed an inverted cone-shaped circulation induction zone downwards.

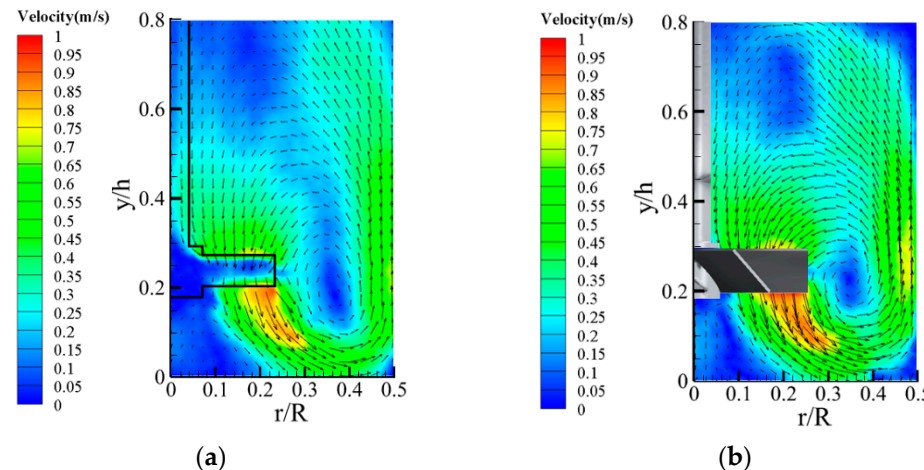

| (a) | (b) |

**Figure 6.** The average velocity field of a fluid. (**a**) The PIV experiment; (**b**) the CFD simulation.

## 5. Results and Discussion

*5.1. Effect of Baffle Structure on Flow Field*

Central agitation would generate a circumferential periodic flow field, and flow state analysis was conducted by extracting the vortex structure in the tank. Figures 7 and 8, respectively, show the flow fields of different relative tooth heights (*A/W*) and relative tooth widths (*λ/h*) on the impeller section. The results show that the circulating fluid would form a flow dead zone before and after the baffle because the baffle would generate a certain flow resistance. As shown in Figures 9 and 10, there was a low-speed region in the range

of r/R = 0.8~0.9. Compared with the standard baffle (r/R = 0.75~0.9), sinusoidal cutting was carried out on the edge of the baffle to make the vortex center close to the wall of the mixing tank, eliminate the small reflux vortex caused by obstruction in front of the baffle, and reduce the reflux flow behind the baffle. A prediction could be made that the reduction of the vortex area would be beneficial for improving the mixing effect and reducing the power consumption.

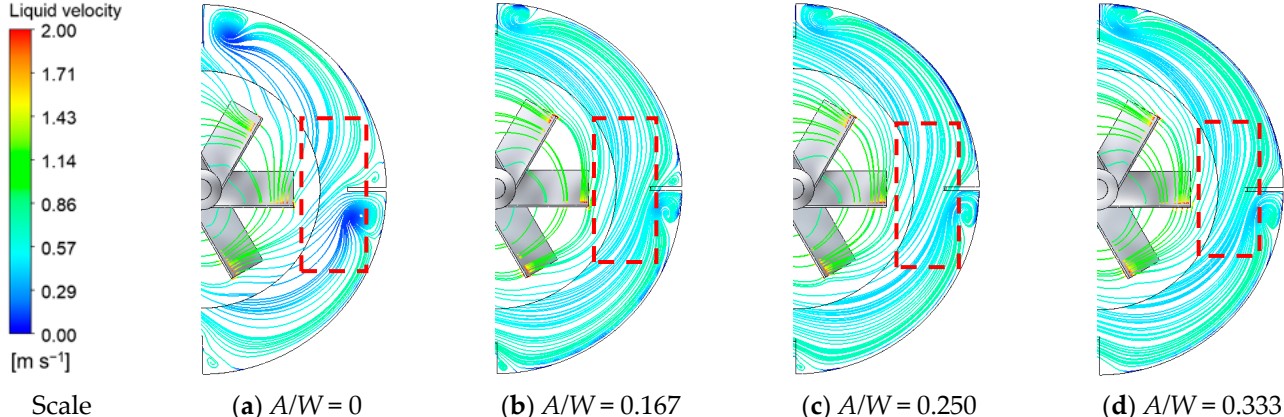

**Figure 7.** The velocity streamline distribution ($y/h$ = 0.25) of the baffle with different relative tooth heights ($A/W$) when $\lambda/h$ = 0.104.

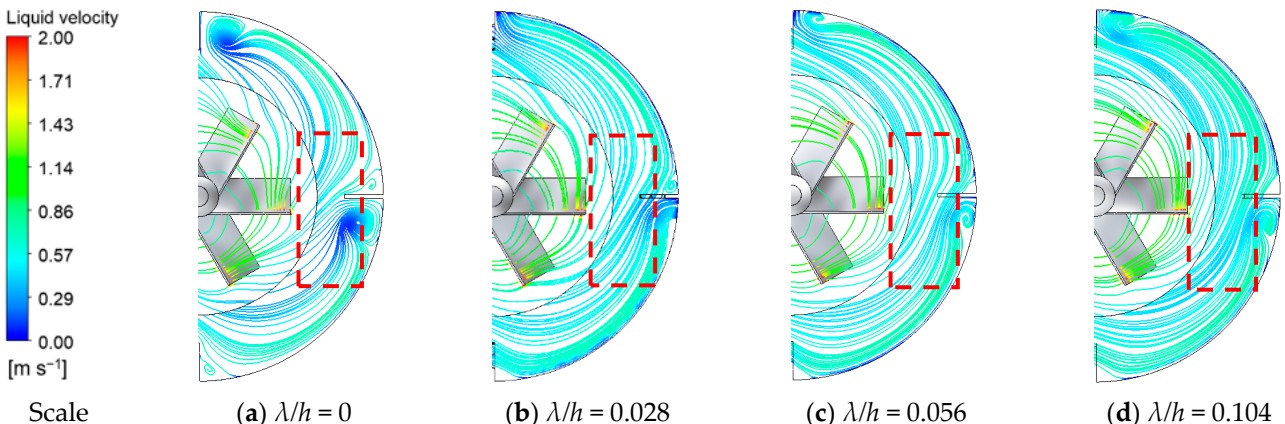

**Figure 8.** The velocity streamline distribution ($y/h$ = 0.25) of the baffle with different relative tooth widths ($\lambda/h$) when $A/W$ = 0.333.

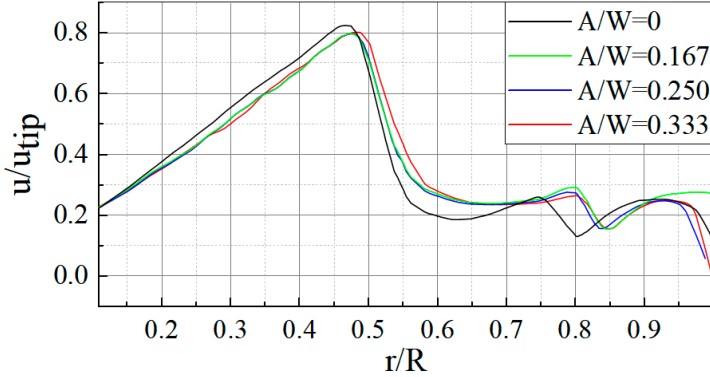

**Figure 9.** A comparison of the average velocity along the radial direction ($y/h$ = 0.25) of the baffle with different relative tooth heights ($A/W$) when $\lambda/h$ = 0.104.

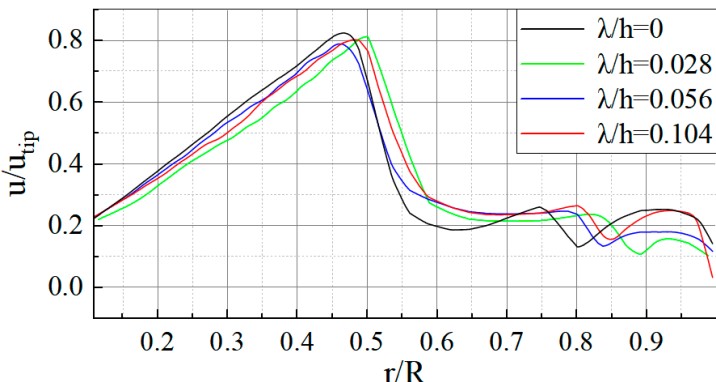

**Figure 10.** A comparison of the average velocity along the radial direction ($y/h$ = 0.25) of the baffle with different relative tooth widths ($\lambda/h$) when $A/W$ = 0.333.

Unlike the serrated trailing edge of the fan, the fluid was relatively perpendicular to the serrated baffle during the stirring process. Limited by the width of the baffle, the tooth height could only be changed in a small range, and thus the change of the relative tooth height ($A/W$) causing the change of the vortex behind the baffle was not obvious. In contrast, the reduction of the tooth width made the backflow behind the baffle tend to weaken. When $A/W$ = 0.333 and $\lambda/h$ = 0.028, the recirculation dead zone behind the baffle was significantly reduced, as shown in Figure 8b.

Figure 11 depicts the radial and tangential velocity distributions of the fluid on the impeller plane. The negative value in Figure 11a can be attributed to the fluid being unable to pass directly through the baffle area for the obstruction of the baffle; thus, a vector was pointing to the axis. The notch on the edge of the sawtooth baffle allowed part of the fluid to pass through the notch to merge with the fluid bypassing the baffle, which weakened the baffle's blocking effect on the fluid and reduced the fluid flow resistance; thus, the negative value of the radial velocity was small. The change of radial velocity enhanced the annular motion of the fluid. In the annular flow domain between the blade and the baffle ($r/R$ = 0.6~0.9), the tangential velocity of the sinusoidal sawtooth baffle was significantly higher than that of the standard baffle.

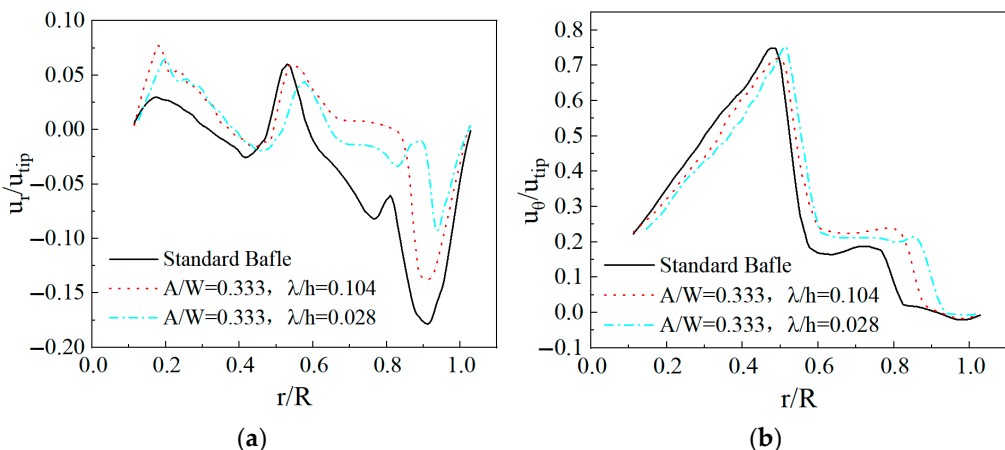

(**a**)　　　　　　　　　　　　　　　(**b**)

**Figure 11.** A velocity comparison ($y/h$ = 0.25) in the annular flow area ($r/R$ = 0.6~0.9) between the impeller and the baffle. (**a**) The radial velocity; (**b**) the tangential velocity.

### 5.2. Effect of Baffle Structure on Flow Pattern behind It

Vortex is a unique form of fluid motion, and the size of the vortex affects the input power. In research on the stirring flow field and power consumption, an investigation into the vortex structure is of considerable significance. During the suspension of solid-phase particles, part of the energy of the impeller was dissipated by the large-scale vortices at

the baffle. In Figures 12 and 13, the three-dimensional structure of the vortex behind the baffle is, respectively, compared in terms of different tooth heights and different tooth widths. The results show that there were large-scale vortices in the lower part of the baffle, which were weakened by the bionic baffle structure and tended to be replaced by weaker eddies. However, the change caused by relative tooth height (*A/W*) was weaker (Figure 12). Compared with the former, the change of the vortex in Figure 13 was more obvious. The relative tooth width (*λ/h*) decreased, that is, the number of teeth increased, so that the larger vortex at the baffle was dispersed into smaller-scale eddies, and the fluid passed through the sawtooth to generate a higher level of turbulence, which could promote material mixing.

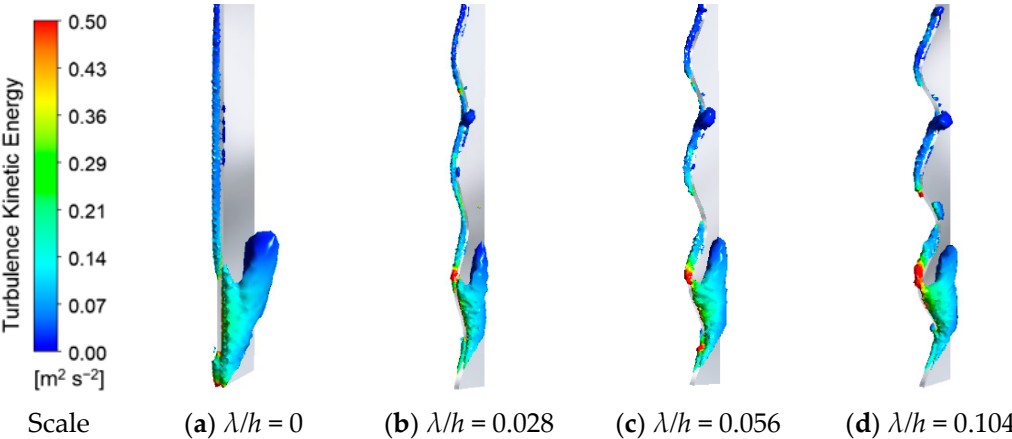

**Figure 12.** The effect of turbulent kinetic energy on the 3D vortex structure behind the baffle at different tooth heights when *λ/h* = 0.104.

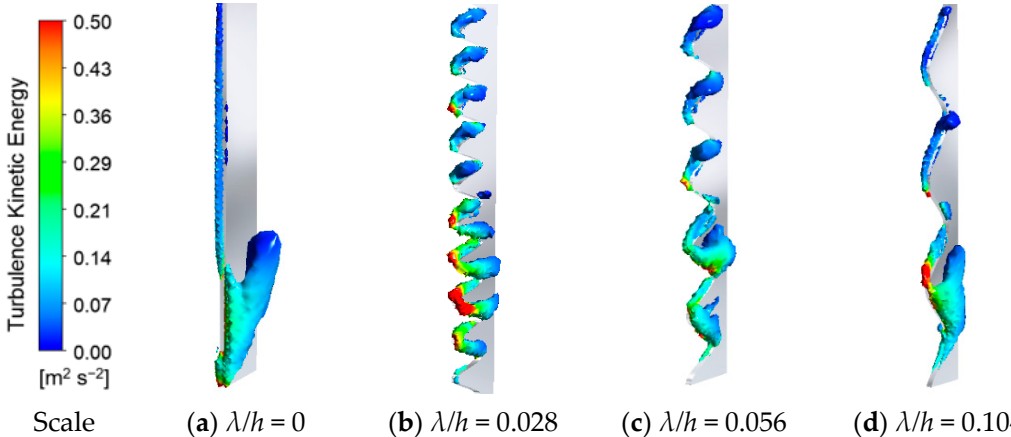

**Figure 13.** The effect of turbulent kinetic energy on the 3D vortex structure behind the baffle at different tooth heights when *A/W* = 0.333.

Figure 14 reflects the fluctuating level of pressure on the backwater side of the baffle. Between *y/h* = 0~0.4, the single broad-scale vortex behind the standard baffle was decomposed by the sawtooth structure. Between *y/h* = 0.4~1, the pressure of the standard baffle did not change, and the fluid disturbance in the middle and upper part of the modified rear baffle was improved. However, the pressure change caused by the increase of the relative tooth height (*A/W*) basically maintained the same fluctuation level. Notably, the smaller the relative tooth width (*λ/h*), the greater the overall pressure fluctuation. When *λ/h* = 0.028, the diverting effect of sawtooth was the most prominent. The turbulence effect of the sawtooth structure effectively broke the large-width low-pressure area at the bottom of the baffle plate and enhanced the pressure gradient, which was beneficial for improving energy utilization and stirring efficiency.

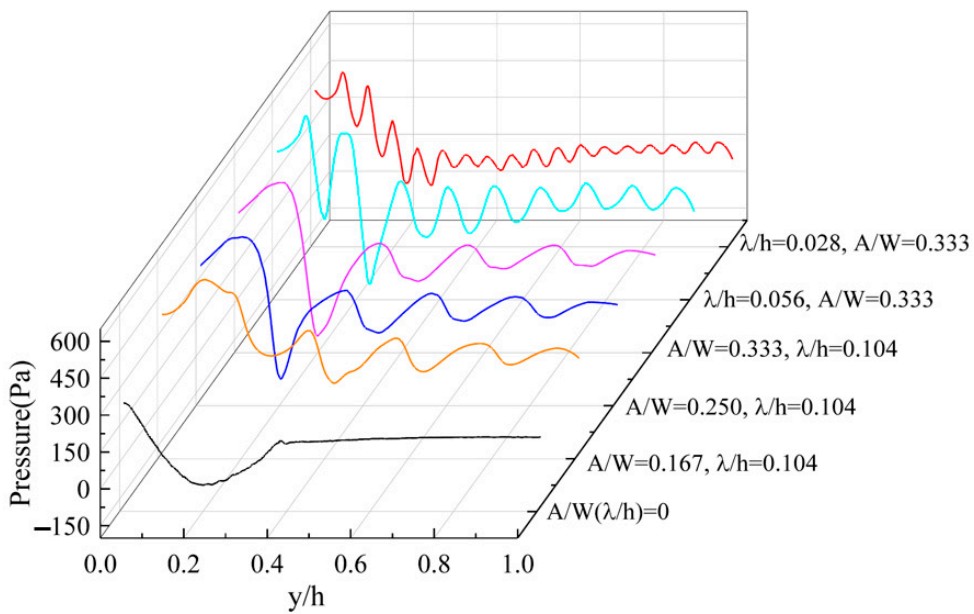

**Figure 14.** The pressure distribution behind the baffle.

## 5.3. Effect of Baffle Structure on Turbulent Dynamics

The turbulent kinetic energy and turbulent dissipation rates affect the mixing time and liquid–solid suspension quality. Figures 15 and 16 depict the distribution of turbulent kinetic energy and the turbulent dissipation rate for different relative tooth heights ($A/W$) and different relative tooth widths ($\lambda/h$), respectively. An observation can be made that under all operating conditions, the turbulent kinetic energy and turbulent dissipation rates had the largest peaks near the blade tip, while the turbulent dissipation rate decreased rapidly outside the impeller region. In the area near the impeller (from $r/R = 0.4\sim r/R = 0.6$), the sawtooth baffles all improved the turbulent kinetic energy of the fluid in the whole vessel. As the relative tooth height ($A/W$) increased and the relative tooth width ($\lambda/h$) decreased, the greater the turbulent kinetic energy of the fluid, the stronger the turbulent diffusion ability. In contrast, the tooth width had a greater influence on the turbulence parameters, indicating that the mixing of the medium was more homogeneous.

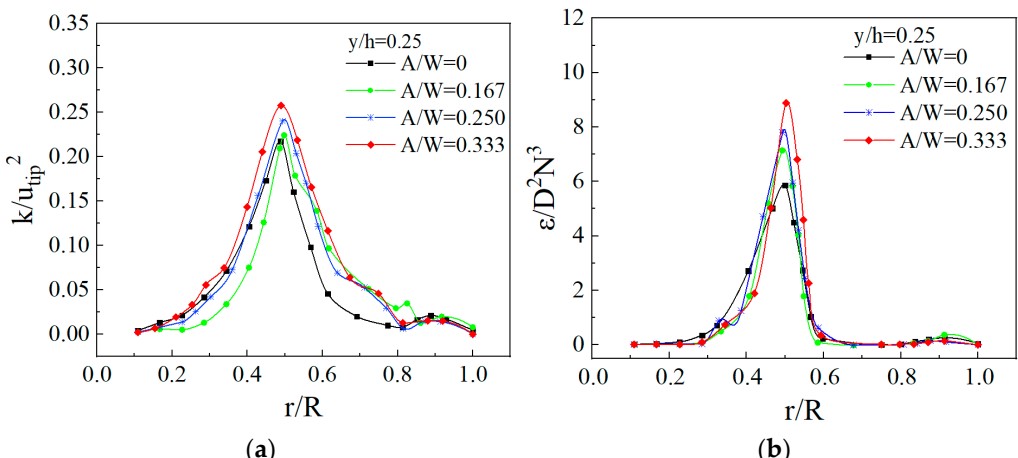

**Figure 15.** The effect of relative tooth height on local radial turbulence dynamics ($\lambda/h = 0.104$). (**a**) The turbulent kinetic energy; (**b**) the turbulent dissipation rate.

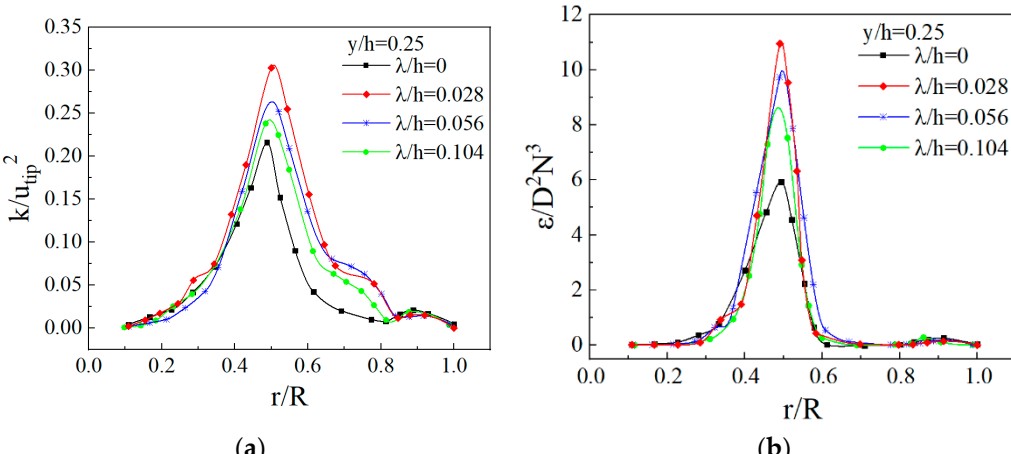

**Figure 16.** The effect of relative tooth width on local radial turbulence dynamics (*A/W* = 0.333). (**a**) The turbulent kinetic energy; (**b**) the turbulent dissipation rate.

### 5.4. Effect of Baffle Structure on Particle Concentration Distribution

In order to explore the uniformity of particle distribution in the flow field, Figure 17 presents the contour plot of particle volume fraction before and after the baffle modification. The sine baffles increased the solid volume fraction in the annular flow region between the impeller and the baffles. The homogenization of solid phase distribution was improved when *A/W* = 0.333, *λ/h* = 0.104 and *A/W* = 0.333, *λ/h* = 0.028, and the latter was more perceptible. However, the fluid tended to move out of the wall, and the particle concentration in the blade gap decreased because of the slotted baffle. Additionally, compared with the standard baffle, the particle uniformity behind the baffle was reduced when *A/W* = 0.333, *λ/h* = 0.104 (0.25 h). Since the sine baffle was still installed vertically on the wall, compared with before modification, the intensification of the sawtooth baffle in the axial direction was not clear, and only the change of the particle flow direction caused by the baffle slot was shown (0.6 h).

Figure 18 depicts the distribution of solid particle concentration in the radial direction of the modified baffle, with the curve showing two peaks. From r/R = 0.4 to r/R = 0.6, the solid particle concentration was higher due to being in the watershed where the impeller blades directly operated. The reason for the peak value at r/R = 0.8~1.0 is that in the typical flow field of the axial-flow impeller, the liquid flowed out of the impeller tip and contacted the wall of tank for diversion, and a large-scale main circulation vortex and inverted conical circulation induction zone were formed at the circumferential direction and bottom of the impeller, respectively (as shown in Figure 6). The annular vortex rolled up the particles and moved upward along the wall of tank, and the sawtooth structure strengthened the degree of turbulence at the baffle, making the peak more obvious. Figure 18 shows that the sinusoidal sawtooth baffle had the beneficial effect of increasing the particle concentration, and between r/R = 0.5~0.8, the volume fraction had a higher level due to the larger fluid velocity after the baffle was cut. When the relative tooth height (*A/W*) increased and the relative tooth width (*λ/h*) decreased, the particle volume fraction increased accordingly. Further, the relative tooth width (*λ/h*) was more prominent than the relative tooth height (*A/W*).

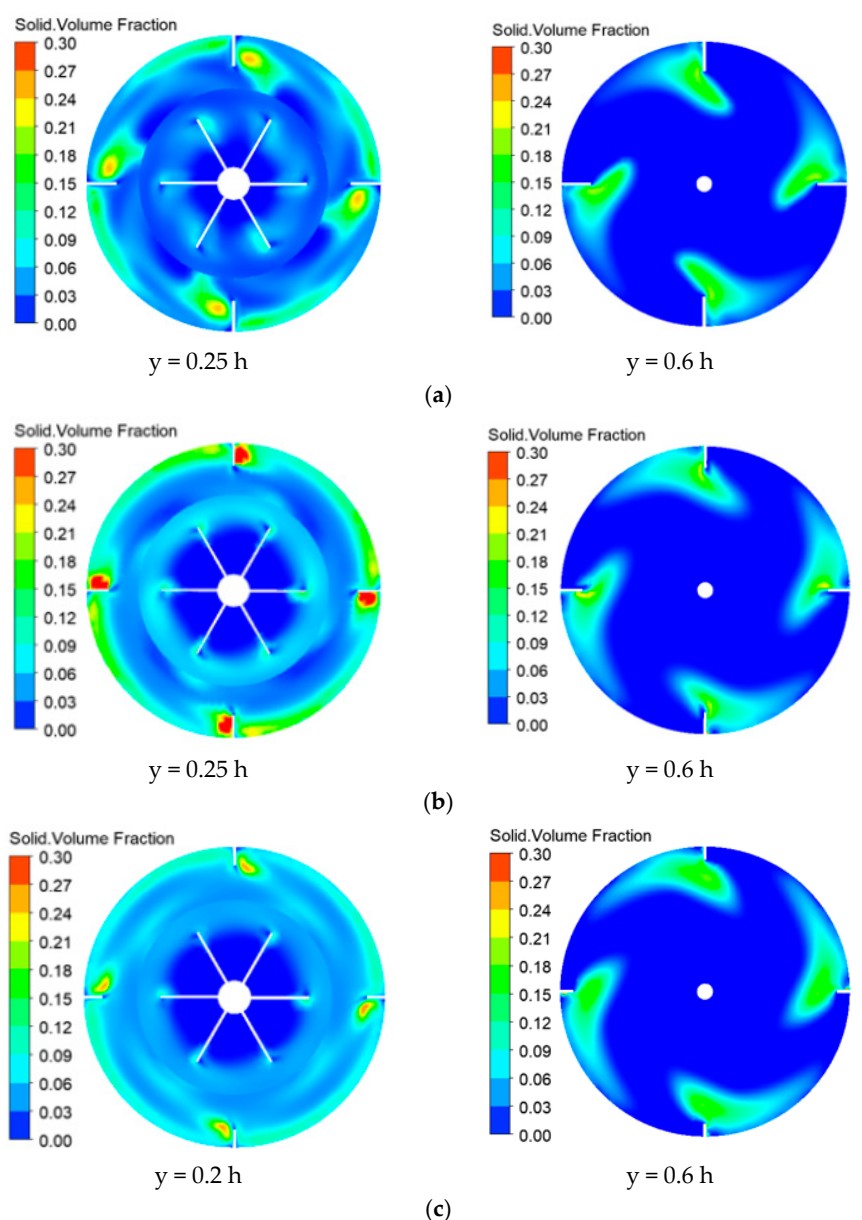

**Figure 17.** The particle volume fraction comparison before and after baffle modification. (**a**) The standard baffle; (**b**) $A/W = 0.333$, $\lambda/h = 0.104$; and (**c**) $A/W = 0.333$, $\lambda/h = 0.028$.

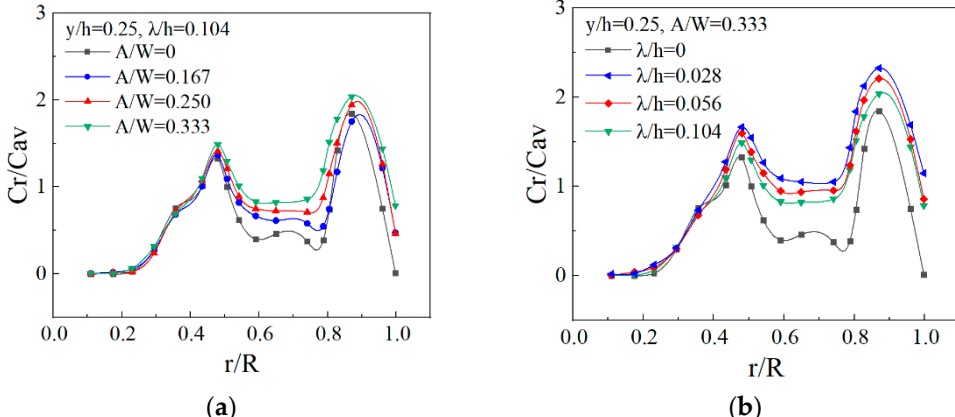

**Figure 18.** The particle concentration distribution in the radial direction. (**a**) The effect of relative tooth height ($A/W$); (**b**) the effect of relative tooth width ($\lambda/h$).

*5.5. Strain Force and Power Consumption*

For stirred tanks in industrial applications, power consumption is a significant cost consideration. At the same rotation speed, the larger the strain rate, the smaller the strain stress (that is, the smaller the fluid resistance received), and the lower the input power requirement for the shaft. Compared with Figure 19a, the modification had little effect on the strain rate near the blade tip, but the baffle cut enhanced the average strain rate. In the annular flow area between the impeller and the baffle, the strain area was enlarged (0.25 h), which was also more conducive to the homogenization distribution of particles. The modified baffle improved the uniformity of the strain rate distribution outside the impeller and was also beneficial to the average degree of the axial strain rate distribution. When *A/W* = 0.333 and *λ/h* = 0.028, the strain rate near the axis was obviously enhanced (0.6 h, 0.95 h), but the strain-rate distributions between different modified structures were considerably similar.

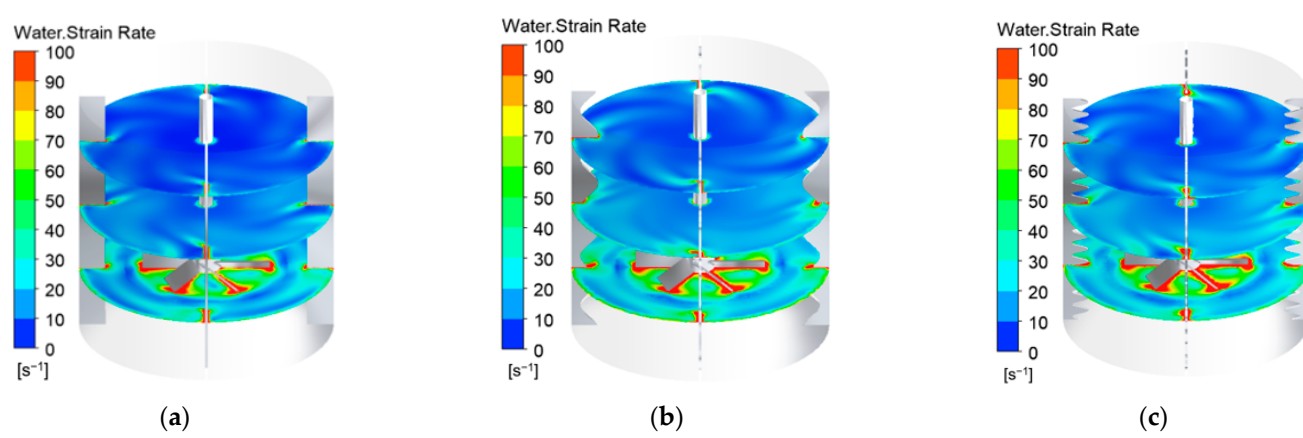

**Figure 19.** The contour diagrams of strain rates at different heights before and after baffle modification (0.25 h, 0.6 h, and 0.95 h). (**a**) The standard baffle; (**b**) *A/W* = 0.333, *λ/h* = 0.104; and (**c**) *A/W* = 0.333, *λ/h* = 0.028.

As shown in Table 4, comparing the power of different tooth heights under *λ/h* = 0.104, with the increase of the sinusoidal sawtooth amplitude, the power consumption decreased more obviously, with a maximum reduction of 9.8%. When *A/W* = 0.333, the relative tooth width decreased, the power consumption tended to be lower, and the power consumption was reduced by 11.7% (Table 5). By analogy with the experimental results of Sivashanmugam et al. [24], the result of the sawtooth baffle for reducing power consumption was reliable.

**Table 4.** The effect of relative tooth height on power consumption.

| Relative Tooth Width (*λ/h*) | Relative Tooth Height (*A/W*) | Power (W) | Reduced Power Consumption (%) |
|---|---|---|---|
| | 0 | 21.4 | - |
| 0.104 | 0.167 | 19.6 | 8.4 |
| | 0.250 | 19.5 | 8.9 |
| | 0.333 | 19.3 | 9.8 |

**Table 5.** The effect of relative tooth width on power consumption.

| Relative Tooth Width (*λ/h*) | Relative Tooth Height (*A/W*) | Power (W) | Reduced Power Consumption (%) |
|---|---|---|---|
| | 0 | 21.4 | - |
| 0.333 | 0.028 | 18.9 | 11.7 |
| | 0.056 | 19.0 | 11.2 |
| | 0.104 | 19.1 | 10.7 |

## 6. Conclusions

In order to improve the stirring performance and reduce the power consumption, a bionic sinusoidal sawtooth baffle was established in the present study. The numerical model was verified by means of the CFD method and PIV experiments, as well as the experimental and computational data of Foukrach [14], Zhang [18], and others. Additionally, a comparative investigation and performance evaluation of several baffles with different tooth heights and tooth widths was conducted.

After the modification, the flow mixing performance of the fluid was improved. The sinusoidal sawtooth baffle reduced the fluid flow resistance, reduced the backflow behind the baffle, and increased the stirring area by grooving the baffle plane. The sawtooth structure broke the large-scale and wide-ranging agglomeration vortices behind the standard baffle, enhanced the degree of turbulence at the baffle, and increased the pressure gradient. The flow mixing was intensified, which was beneficial for improving the energy transfer and stirring efficiency. With the decrease of the relative tooth width ($\lambda/h$) and the increase of the relative tooth height ($A/W$), the turbulent flow parameters in the container tended to increase. To a certain extent, the speed in the annular flow area between the baffle and the impeller (r/R = 0.6~0.9) was ameliorated, the suspended quality and distribution of particles were improved, and the strain rate outside and in the axial direction of the impeller was strengthened. Unlike the change caused by the increase of the relative tooth height ($A/W$), the decrease of the relative tooth width ($\lambda/h$) had a more prominent effect on the flow field structure and power consumption. When $A/W$ = 0.333 and $\lambda/h$ = 0.028, the maximum power consumption was reduced by about 11.7%.

**Author Contributions:** Methodology, Q.Y.; validation, W.Z. and L.L.; investigation, L.L. and D.X.; data curation, S.Z.; writing—original draft preparation, Q.Y.; writing—review and editing, S.Z.; project administration, S.Z.; and funding acquisition, S.Z. and H.Y. All authors have read and agreed to the published version of the manuscript.

**Funding:** This research was funded by National Natural Science Foundation of China, grant number 51706203; National Science and Technology Major Special Sub Project, grant number 2019zx06004001; Natural Science Foundation of Zhejiang Province, Exploration Project, grant number Y, LY20E090004; and the Fundamental Research Funds for the Central Universities, grant number JZ2021HGB0090.

**Institutional Review Board Statement:** Not applicable.

**Informed Consent Statement:** Not applicable.

**Data Availability Statement:** Data available on request due to restrictions eg privacy or ethical. The data presented in this study are available on request from the corresponding author.

**Acknowledgments:** Thanks to Zhejiang University of Technology for providing computing resources and technical support. The authors also appreciate all other scholars for their advice and assistance in improving this article.

**Conflicts of Interest:** The authors declare no conflict of interest.

## Nomenclature

| | |
|---|---|
| $A$ | Tooth height, mm |
| $C$ | Off-bottom clearance, mm |
| $C_{av}$ | Average concentration |
| $C_d$ | Drag coefficient |
| $C_r$ | Radial concentration |
| $C_{1\varepsilon}, C_{2\varepsilon}, C_\mu$ | Coefficients of turbulent model |
| $D$ | Impeller diameter, mm |
| $d_S$ | Solid particle diameter, mm |
| $G_{k,m}$ | Turbulence generation, $\text{kg·m/s}^3$ |
| $g$ | Gravity acceleration, $\text{m/s}^2$ |
| $h$ | Baffle height, mm |
| $K$ | Interface momentum transfer coefficient |

| | |
|---|---|
| $k$ | Turbulent kinetic energy, $m^2/s^2$ |
| $N$ | Impeller speed, rpm |
| $P$ | Power, W |
| $p$ | Pressure, pa |
| $Re$ | Reynolds number |
| $R$ | Radius of the tank, mm |
| $T$ | Tank diameter, mm |
| $t$ | Time, s |
| $\vec{u}$ | Velocity vector, m/s |
| $u$ | Average velocity, m/s |
| $u_r$, $u_\theta$, | cylindrical velocity components, m/s |
| $u_{tip}$ | Impeller tip velocity, m/s |
| $W$ | Baffle width, mm |
| $W_b$ | Blade width, mm |
| $x,y,z$ | Cartesian coordinates |
| Greek Letters | |
| $\lambda$ | Tooth width, mm |
| $\delta$ | Baffle thickness, mm |
| $\varepsilon$ | Turbulent kinetic energy dissipation rate, $m^2/s^3$ |
| $\mu$ | Shear viscosity, Pa·s |
| $\rho$ | Density, $kg/m^3$ |
| $\sigma$ | Turbulent Prandtl number for $k$ and $\varepsilon$ |
| $\bar{\bar{\tau}}$ | Viscous stress tensor, Pa |
| $\varphi$ | Volume fraction, % |
| $\psi$ | Smooth function |
| Subscripts | |
| $L$ | Liquid phase |
| $S$ | Solid phase |
| $m$ | Mixture properties |
| $i$ | $i$ = L for liquid phase or S for solid phase |

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
