# Peer review of "CFD Analysis of Sine Baffles on Flow Mixing and Power Consumption in Stirred Tank"

_applsci, doi:10.3390/app12115743_

Round 1
Reviewer 1 Report
Please see attached file.

Author Response
Dear Reviewer,
Thank you for your comments on the manuscript applsci-1741047. We have revised it according to your comments. Please see the attachment.
Sincerely,
Qizhi Yang

Reviewer 2 Report
The topic is interesting. Detailed flow studies in spatially resolved CFD and experiments are a matter of interest. However, there are some points that should be addressed before considering the paper for publication. I list them below:
- As a general comment, please be more accurate when using some terms. Many statements in the paper are imprecise or used in a very uncommon way.
- Please check the term "bionic" in the title and the manuscript. I believe that it is misleading, provided that, formally speaking, bionic is a term that specifically refers to body parts with artificial/mechanical parts. Maybe you wanted to say biomimetic or so. But, in my opinion, this is more like sinusoidal or waved baffles. Please correct the title and the term along with the manuscript.
- In the abstract, please, also replace the A/W and lambda/h for more meaningful. At that point, such variables haven't been defined, so please use something like the ratio between... or so.
- Line 61: what do you mean by "turbulence degree"? Turbulence is a flow regime. Maybe you are referring to turbulence intensity or something like that.
- Correct the format of the in-text reference.
- Line 199: What do you mean by "upwind style"? maybe you wanted to say upwind scheme.
- I strongly recommend sending the paper for English spelling checking. Some sentences are very uncommon. for example:
- Line 238: The stirring flow field was a typical periodic flow field
- Line 241: vortices were formed at the baffles due to the obstruction of the baffles
- Line 243: What do you mean by "eddy current"? Even so, if this is something related to the flow you may want to say stream instead of current.
- title of section 5.2: First, please avoid using eddy current. You may use pattern or something like that instead of current. Second, eddy is a very specific term referring to turbulent vortices, that are part of the vortex energy cascade, which is not the case that you are analyzing.
- Section 5.4: It seems that the analysis is purely qualitative. Please add some quantitative analysis, for example, the mean and standard deviation of the particle distribution.
- Line 406: What does it mean "the overall flow performance of the fluid was improved"? Please clarify.
- Figure 6: the y-axis label says speed. It should be velocity.
- The maximum CFL number is missing. Please report.
- The time discretization scheme is missing. Please add.
- The time-step stop criterion is missing. Please add.
- The wall treatment is missing in the text. Please add.
- The maximum wall y+ is missing. Please report.
Author Response

(The authors gave the same response as above.)

Reviewer 3 Report
This paper is valuable considering the research about the baffle design in mixing process. The subject is well introduced and developed. The main points concerning the mixing are analysed. The writing form is suitable. But some typos must be corrected before publishing. In particular:
*) the list of references must be checked and the citing expressions in the text must be corrected;
*) the simulation conditions must be fully mentioned (A/W and Lambda/h) for the analysis of the results;
*) the presentation of the mathematical model, in Section 4, must be more precise. Each new term must be presented.
After the minor corrections listed in the attached file, I recommend the publication.

Author Response

(The authors gave the same response as above.)

Round 2
Reviewer 2 Report
Most of my previous comments were addressed. However, I still see many "Error! Reference source not found" messages in the pdf file that I received. Please correct that. Check also that there are many variables with the units missing in the nomenclature table.
Author Response
Dear Reviewer,
Thank you for your comments on the manuscript applsci-1741047. We have revised it according to your comments. Attached is the PDF format of the revised manuscript
Sincerely,
Qizhi Yang
